# One-Class SVM on siamese neural network latent space for Unsupervised Anomaly Detection on brain MRI White Matter Hyperintensities

**Nicolas Pinon**[1]                                    NICOLAS.PINON@CREATIS.INSA-LYON.FR
**Robin Trombetta**[1]                              ROBIN.TROMBETTA@CREATIS.INSA-LYON.FR
**Carole Lartizien**[1]                            CAROLE.LARTIZIEN@CREATIS.INSA-LYON.FR
[1] *Univ. Lyon, CNRS UMR 5220, Inserm U1294, INSA Lyon, UCBL, CREATIS, France*

**Editors:** Accepted for publication at MIDL 2023

## Abstract

Anomaly detection remains a challenging task in neuroimaging when little to no supervision is available and when lesions can be very small or with subtle contrast. Patch-based representation learning has shown powerful representation capacities when applied to industrial or medical imaging and outlier detection methods have been applied successfully to these images. In this work, we propose an unsupervised anomaly detection (UAD) method based on a latent space constructed by a siamese patch-based auto-encoder and perform the outlier detection with a One-Class SVM training paradigm tailored to the lesion detection task in multi-modality neuroimaging. We evaluate performances of this model on a public database, the White Matter Hyperintensities (WMH) challenge and show in par performance with the two best performing state-of-the-art methods reported so far.

**Keywords:** Anomaly detection, One-Class SVM, Auto-encoder, Representation learning, Brain MRI, White Matter Hyperintensities

## 1. Introduction

*Unsupervised Anomaly Detection* (UAD), also referred to as *outlier detection*, has been proposed as an alternative to deep supervised learning for medical image analysis when the studied pathology is either rare or with heterogeneous patterns as well as when getting labels from radiologists is very challenging. This formalism relies on the estimation of the distribution of normal (i.e. non-pathological) data in some representation space associated to some distance metric allowing to quantify the deviation of test samples from this normal model at inference time. Samples highly deviating from the normal distribution are referred to as *outliers*.

Different categories of UAD methods have been recently applied to medical image segmentation or detection tasks. The most popular formalism is based on auto-encoders (AE) architectures that are trained on normal images only to perform a "pretext" task consisting in the reconstruction of the input image. During inference, voxel-wise anomaly scores are computed in the image space as the reconstruction errors, *i.e.* the differences between the input test image and the pseudo-normal reconstructed one, assuming that the AE has initially well captured the normal subjects main features and will thus generate large errors for anomalous voxels contained in the test image. Such models have successfully been applied to the *segmentation* of visible brain anomalies in MRI medical image analysis (Baur

et al., 2021). Recent observations, however, outline the limitations of the reconstruction error scores for the detection of very subtle abnormalities (Meissen et al., 2022). Different attempts have been proposed to overcome the reported limitations of these approaches. Pinaya et al proposed an advanced architecture combining a vector quantized auto-encoder (VQ-VAE) with autoregressive transformers acting in the latent representation space to explicitly model the likelihood function of the discrete elements (Pinaya et al., 2022). This was shown to perform well in different neuroimaging applications including the MICCAI WMH challenge that tackles the detection of white matter hyperintensity (WMH) lesions in T1 and FLAIR MRI. However, the sequential nature of auto-regressive models introduces bias, which can be handled with an ensemble of models but at the cost of a much higher computation time at inference. Alaverdyan *et al.* proposed to perform the detection step in a latent space by coupling the representation power of patch-based AE networks to extract relevant and subtle features with the efficiency of a multivariate non parametric discriminative one class support vector machine (OC-SVM) (Alaverdyan et al., 2020). This model was shown to achieve promising results for the detection of subtle (MRI negative) epilepsy lesions in T1 and FLAIR MRI. In this work, we build on the formalism proposed in (Alaverdyan et al., 2020) and propose an original discriminative model that is compared to the state-of-the art generative architecture proposed by (Pinaya et al., 2022) and (Baur et al., 2021). Performance of these models, trained on the same database of normal control exams (Mérida et al., 2020), are evaluated on the WMH challenge T1 and FLAIR MRI data to guarantee a fair comparison.

## 2. Proposed UAD method

The proposed UAD pipeline is depicted on figure 1. It consists of two main steps, the **representation learning** step which constructs a latent space of the distribution of normal samples based on an auto-encoder trained on patches extracted from a control population. The **outlier detection** step which estimates the support of the normality distribution for each patient based on a one-class support vector machine (OC-SVM) trained on a subset of patches extracted from the patient under consideration.

### 2.1. Representation learning

With the goal of facilitating outlier detection, we first construct a representation space of latent variable $\mathbf{z}$ that is suitable for this task. Leveraging the architecture proposed in (Alaverdyan et al., 2020), we train a siamese auto-encoder (SAE) to reconstruct 2D patches of the input image, where the patch-based approach is aimed at enriching the latent space with local information at the voxel level. The siamese auto-encoder is composed of two auto-encoders with shared weights, each taking a couple of input patches $(\mathbf{x_1}, \mathbf{x_2})$ from different subjects, but located at the same location in the brain. The patches are drawn from a set of normal (i.e. healthy) images $\mathcal{X} = \mathcal{X}_{1 \leq h \leq N_{\mathrm{hc}}}^h = (\mathbf{x}_i^h)_{1 \leq i \leq m, 1 \leq h \leq N_{\mathrm{hc}}}$ with $m$ the number of locations and $N_{\mathrm{hc}}$ the number of healthy controls. The network first encodes the patches $(\mathbf{x_1}, \mathbf{x_2})$ into latent representations $(\mathbf{z_1}, \mathbf{z_2})$ and decodes them into reconstructions $(\hat{\mathbf{x}}_1, \hat{\mathbf{x}}_2)$. While training this network has to balance two terms for the loss 1) a cosine similarity term, ensuring that the couple of patches located in the same location in the brain are close when projected in the latent space, thus constructing a meaningful representation space 2)

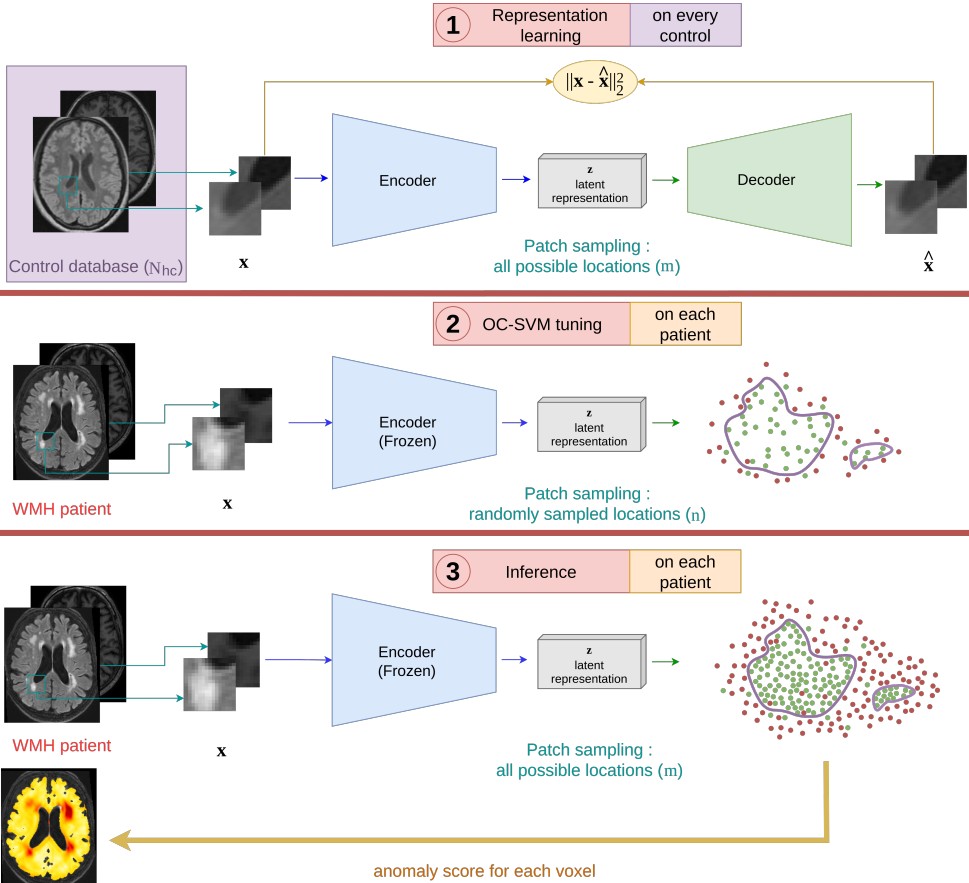

Figure 1: Proposed UAD pipeline consisting of 3 steps : 1) representation learning on the whole control database 2) OC-SVM tuning on each patient using a subset of latent representations $\mathbf{z}$ 3) inference on the whole brain using previously tuned OC-SVM.

a reconstruction error term between the original couple of patches and their reconstructions ensuring that the encoder does not collapse to a single representation.

$$L_{SAE}(\mathbf{x_1}, \mathbf{x_2}) = \sum_{t=1}^{2} ||\mathbf{x_t} - \hat{\mathbf{x}}_\mathbf{t}||_2^2 - \alpha \cdot cos(\mathbf{z_1}, \mathbf{z_2})$$

The training is done on patches of the input images, with the goal of constructing an encoder that captures fine and subtle information. Patches are extracted from healthy controls only with the assumption that the latent variable $\mathbf{z}$ extracted from pathological area will appear as out of distribution in the inference stage. At the end of the training, the encoder's weights are frozen and used to extract latent representation $\mathbf{z}$ from any image patch $\mathbf{x}$.

## 2.2. Outlier detection

We propose to perform the outlier detection step in the latent space of the auto-encoder by estimating the support of the normal (i.e. healthy) probability distribution. To this end, we

use a one-class support vector machine (OC-SVM) algorithm (Schölkopf and Smola, 2002) whose goal is to construct a decision function $f$ that is positive on the estimated normal distribution support and negative outside. Samples $\mathbf{z}_i$ are projected to a high dimensional space with a mapping $\phi(\cdot)$ associated with a kernel $k$ such that $k(\mathbf{z}_i, \mathbf{z}_j) = \phi(\mathbf{z}_i) \cdot \phi(\mathbf{z}_j)$. The kernel is chosen to be the radial basis function kernel, i.e. $k(\mathbf{z}_i, \mathbf{z}_j) = \exp(-\gamma ||\mathbf{z}_i - \mathbf{z}_j||_2^2)$. This guarantees that the problem is linear in this redescription space so that the decision function is a hyperplane of equation $\boldsymbol{w} \cdot \boldsymbol{\phi}(\mathbf{z}) - \rho = 0$. An additional parameter $\nu$ adjusts the upper bound on the fraction of permitted training errors, allowing to account for the presence of "non-normal" samples in the training samples.

Based on this $\nu$-property and assuming that the lesion area represents a negligible fraction of patient images, we propose to build one normality model per patient $p$, by randomly sampling $n$ voxel locations throughout the $m$ brain voxel locations with $n \ll m$. Patches centered on these $n$ locations are fed through the SAE network to constitute a subset $\mathcal{Z}^p = (\mathbf{z}_i^p)_{1 \leq i \leq n}$ that serves to estimate the decision function $f_p$ of parameters $\mathbf{w}_p$ and $\rho_p$ for patient $p$. Once the decision function is computed, we can infer an outlier score for each of the $m$ voxels of the patient image, by extracting the patch centered on this voxel, feeding it then to the SAE to derive its latent representation $\mathbf{z}$, which is in turn inputted to the decision function $f_p$ to estimate its distance to the normal distribution support. This allows deriving an anomaly score map for the whole brain of patient $p$.

Note that, unlike in (Alaverdyan et al., 2020), this whole outlier detection step is done only at inference stage on each individual patient image and does not need any training on the normal training set $\mathcal{X}$. We hypothesize that computing the normal distribution at patient level will enable detection performance gain as it allows to be independent of the registration quality in this step and also can capture some of the particularity of the patient normal distribution (e.g. large occipital lobe volume, brain shrinkage, etc.).

### 2.3. Post-processing of the anomaly score maps

We found in early experiments that the method was generating a high number of false positives in the cerebrospinal fluid (CSF), whether near the border of the cortex or in the ventricles, as we believe these regions are highly affected by the quality of the registration. To mitigate this effect we use the FMRIB's Automated Segmentation Tool (FAST) by (Zhang et al., 2001) to segment the grey and white matter, allowing us to exclude the CSF from the anomaly maps. Details of this post-processing can be found in appendix E.

### 3. Experiments

### 3.1. Database and splitting

Two different databases are used in this study. **The control dataset** consists of a series of 75 paired T1 weighted and FLAIR MRI scans acquired on healthy volunteers on a 1.5T Siemens Sonata scanner and mCT PET scanner (Siemens Healthcare, Erlangen, Germany). This database was approved by our institutional review board with approval numbers X and X (blinded for review). It serves to learn the normality distribution of the UAD models. **The WMH Segmentation Challenge training**[1] **dataset** contains 60 paired T1 weighted and FLAIR images each associated to the corresponding 3D lesion mask image.

---

1. Note that the test subset, which contains 110 exams, was not available at the time of this study.

These data were acquired on 3 scanners of different vendors in 3 different hospitals in the Netherlands and Singapore, namely 20 exams acquired on a 3 T Philips Achieva of UMC Utrecht, 20 exams exams acquired on a 3 T Siemens TrioTim of NUHS Singapore and 20 exams acquired on a 3 T GE Signa HDxt of VU Amsterdam. All 3D images of the 2 databases were co-registered to the MNI space based on SPM12 processing tools, thus leading to 3D volumes of size 157x189x136 with 1mm$^3$ isotropic voxel size.

### 3.2. Hyper-parameters of the UAD method

The encoder was composed of 4 convolutionnal blocks with the following characteristics : kernel size of $(5, 5)$, $(3, 3)$, $(3, 3)$ and $(3, 3)$, strides, respectively, of $(1, 1)$, $(1, 1)$, $(3, 3)$ and $(1, 1)$, number of filters, respectively, of 3, 4, 12 and 16, no padding and GeLu activation. Each block was followed by a batch normalization block. The decoder was the symmetric counterpart of the encoder, except the last block which did not use batch normalization and used sigmoid activation function. The input of the encoder consisted of patches of each of the 2 modalities combined as channels. The siamese network was trained with 18 750 000 patches of size $15 \times 15 \times 2$ (250 000 patches per subject). We used Adam optimizer (Kingma and Ba, 2015) for 30 epochs, with default hyperparameters, best model selection based on validation loss and training batch size of 1000. For the OC-SVM, $n$ was set to 500 (sampling ratio $\frac{n}{m} \simeq 0.02\%$). We used $\nu = 0.03$ and the hyperparameter $\gamma$ was set such that $\frac{1}{\gamma}$ was equal to the product of variance and dimension of the $\mathbf{z}_i$.

### 3.3. Comparison to other detection methods

We compare our UAD pipeline to what is, to our knowledge, the best performing unsupervised models for anomaly detection and segmentation on the WMH challenge dataset (Baur et al., 2021; Pinaya et al., 2022).

**VQVAE and Transformer** Pinaya et al proposed to combine the discrete latent representation of a Vector Quantized Variational Auto-Encoder (VQ-VAE) with a Transformer (Vaswani et al., 2017) to perform anomaly detection by restoration of lesions on MRI images. The VQ-VAE (van den Oord et al., 2017) is trained to reconstruct whole 2D transverse slices and learn a meaningful discrete latent representation vector of the data. In a second step, the Transformer serves as an autoregressive model to learn the sequences of indices of the quantized latent vectors on the "normal" images only. For each element of the sequence, if the probability of the encoding vector predicted by the autoregressive model is lower than an arbitrary value, this vector is resampled. The new restored sequence then feeds the decoder of the VQ-VAE to get the final image where it is assumed that the anomalies have been "healed". The error map between this output image and the original one is interpreted as an anomaly score to segment the lesions in the brain. An additional step upsamples the mask of the elements that have been resampled by the Transformer and uses it as a mask to filter the residuals maps.

We adapt this approach to our context of experimentation, using the control dataset $\mathcal{X}$ to train the models. The slices were padded to a size of 192x192 and we used the same architectures for the VQ-VAE, resulting in a latent space of size 24x24x256. The same data augmentation and learning parameters are used and the model was trained for 500 epochs with a batch size of 128. During our experimentations, we found that using Performer

(Choromanski et al., 2020) was less efficient and accurate than the regular Transformer, so we chose the latter. It is composed of 8 layers, embedding size of 256 and was train trained for 200 epochs with a batch size of 32, learning rate of 1e-4. The threshold for resampling encoding vectors was set to 0.005. For computational limitations, the improvement induced by the multiplication of pathways in latent space for the Transformer was not evaluated, we only report performance achieved when using the basic raster order.

**Auto-encoder** The performances are also compared to the auto-encoder of (Baur et al., 2021), which has the particularity to include two skip connections in the middle of the network. Anomaly maps are obtained after applying a 5x5x5 median filter on the error maps between the input and the associated reconstruction. The same data augmentation – small random translation, contrast and intensity shifts – is applied on the padded slices and the model is trained during 500 epochs with Adam optimizer with default hyperparameters and a batch size of 128.

**Multiple OC-SVM** To assess the benefits of our patient-specific OC-SVM approach, we use the latent representation obtained with a SAE on the control dataset to tune one OC-SVM per voxel as in (Alaverdyan et al., 2020).

### 3.4. Evaluation methodology

For each method presented, we obtain an anomaly score map, whether it is the score of the decision function of OC-SVM, the restoration error for the VQ-VAE + transformer or the reconstruction error for the AE. We evaluate classical pixel-level metrics : the area under the ROC curve (**AU ROC**), as well as the area under the precision recall curve (**AU PRC**), which intrinsically takes into account large class imbalance and does not use the false positive rate. We also compute the AU ROC corresponding to false positives rates lower than 30% (**AUC ROC 30**) as in (Bergmann et al., 2021). Anomaly maps outlining 30% or more of FP voxels are indeed degenerated and of no use in the medical imaging context where the anomalies take only a small portion of the total volume (0.35% in the WMH database). The small lesional fraction also justifies the use of the $\nu$-property used in section 2.2. As in (Bergmann et al., 2021), we also investigate the area under the Per Region Overlap curve (**AU PRO**), which reports variations of the average PRO as a function of the FP rate, where the PRO is defined as the sensitivity of each individual lesion, and the average PRO is the average values over all lesions in the patient. The PRO acts as a true positive rate (sensitivity) normalized per lesion size, meaning a correctly detected small lesion will count as much as a large one. This metric is particularly relevant in the context of anomaly detection in medical imaging, where we want to detect every lesions regardless of the size, unlike AU ROC which is biased towards detecting large lesions. Finally, as a mean of comparison to (Pinaya et al., 2022), we also investigate the best achievable **Dice**, which is simply the maximum value of the Dice metric obtained with the optimal threshold applied on the score maps. Non parametric Kruskall-Wallis analysis of variance followed by Dunn's tests for multiple comparisons were performed to compare the different metrics.

## 4. Results

Visual assessment of the anomaly maps achieved by the UAD models is showcased on Figure 2 and Appendix C. Table 1 reports performance achieved by our proposed SAE+OC-SVM method with and without accounting for detections in the CSF (see section 2.3)

| 3 hospitals | VQ-VAE + Transformer (Pinaya) | AE (Baur) | SAE + multiple OC-SVM (Alaverdyan) | SAE + OC-SVM (Ours) | SAE +OC-SVM +CSF seg (Ours) |
|---|---|---|---|---|---|
| AU ROC | 0.69 ± 0.13 | 0.53 ± 0.09 | 0.52 ± 0.19 | **0.80** ± 0.09 | **0.81** ± 0.10 |
| AU ROC 30 | 0.40 ± 0.20 | 0.20 ± 0.12 | 0.19 ± 0.16 | **0.48** ± 0.20 | **0.59** ± 0.17 |
| AU PRC | 0.065 ± 0.079 | 0.028 ± 0.030 | 0.023 ± 0.031 | **0.084** ± 0.099 | **0.165** ± 0.168 |
| AU PRO | 0.55 ± 0.10 | 0.50 ± 0.08 | 0.43 ± 0.17 | 0.71 ± 0.11 | **0.80** ± 0.07 |
| AU PRO 30 | 0.19 ± 0.13 | 0.15 ± 0.07 | 0.09 ± 0.13 | 0.33 ± 0.18 | **0.48** ± 0.13 |
| ⌈ Dice ⌉ | 0.11 ± 0.10 | 0.06 ± 0.05 | 0.05 ± 0.05 | **0.14** ± 0.13 | **0.22** ± 0.17 |

Table 1: Mean metric on every patient from the 3 different hospitals for each method. In bold are shown the best model and those for which the statistical difference with the best model for Dunn's test is not significative (p-value ≥ 0.01).

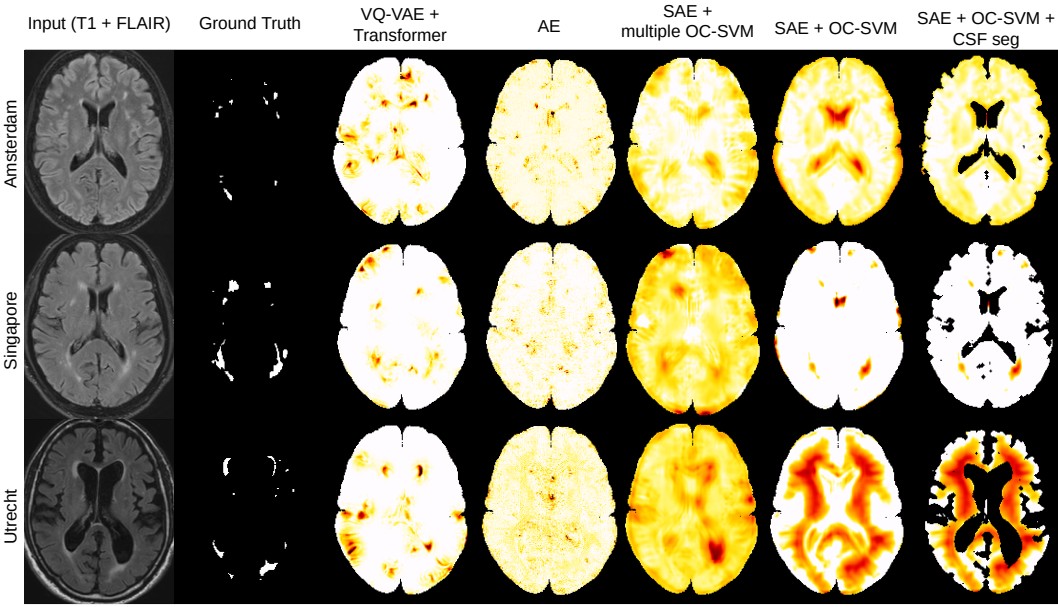

Figure 2: Showcase of the different methods studied on 3 examples from the 3 hospitals

compared to the methods presented in 3.3. Reported metrics correspond to the mean values and standard deviations computed over the pooled 60 exams of the WMH training database. On overall, performance of the SAE+ OC-SVM are better than all three other methods for all the metrics.

As explained in section 2.3, we emphasize that the PRO metric, widely used in the computer vision domain, is well-suited for the evaluation of outlier detection tasks. The good PRO performance achieved with our model is thus a positive indicator of its capacity to detect subtle brain anomalies.

Results of Table 1 show the benefits of tuning one OC-SVM per patient instead of one per voxel, as it significantly improves the performances compared to (Alaverdyan et al., 2020). They also indicate that both our method and the VQ-VAE+Transformer model outperform the simple AE model proposed by (Baur et al., 2021). The two last columns of

Table 1 underline the impact of the proposed post-processing on the SAE+OC-SVM model (on Figure 2, 5th column, we see that our method is likely to generate false detections in the CSF, especially in the ventricles). Reported performance are on overall higher when applying the CSF mask, this is mainly due to a significant reduction of the false positive rate that is not counterbalanced by the paired slight decrease in sensitivity due to the imperfect segmentation of the ventricles encompassing some true lesions located on its border. However, most of the statistical tests did not conclude that the observed differences were significant. To further investigate the comparative performance of the 3 UAD models, we report results achieved by each of the 3 hospitals (Amsterdam, Singapore and Utrecht) in Tables 2, 3, 4, of Appendix A. This study is, as far as we know, the first to report such a detailed analysis by institution. It shows that all the models see their performance decrease on the Utrecht dataset. This could be due to a greater difference between the images in this dataset and those of the control dataset, especially for FLAIR images, highlighting the need to have more heterogeneous controls during the representation learning step to build more robust models. Computation of the global mean lesional volume of each center, however, indicates a dataset shift with mean values of 13495, 26123 and 29296 $mm^3$ for the Amsterdam, Singapore and Utrecht centers, respectively. One possible explanation might thus be that our method does not perform well when data are characterized by a low lesional load. This requires further investigation.

## 5. Discussion and Conclusion

The reported DICE-score and AUPRC in (Pinaya et al., 2022) (Dice = 0.269, AUPRC = 0.158) and (Baur et al., 2021) (Dice=0.45 and AUPRC = 0.37) are much higher than the values reported in this study. However, they do not compare since they were achieved based on UAD models trained on FLAIR data only while we accounted for both T1 and FLAIR (concatenated as 2 input channels) in this study. The significant performance gap is also likely to be explained by methodological differences among the studies. As an example, training of the VQ-VAE + transformer models in (Pinaya et al., 2022) is based on a selection of more than 15 000 FLAIR "normal" volumes from the UK Biobank database, some of them containing WMH lesions. This training dataset, constituted of the 4 central slices of each volume, is thus highly different from ours based on 75 3D volumes of healthy controls (including all transverse slices) who do not contain any WMH lesion. Training of the Pinaya and Baur UAD models in this study included data augmentation focused on intensity scaling and contrast adjustment, unlike ours. Such a strategy may help accounting for the distribution shifts observed among images acquired on the 4 different scanners as reported in Figure 3. As stated in the Introduction, our main purpose was to provide a fair comparison between the different UAD models by training and testing these models on the same datasets as well as using the same evaluation metrics and strategy. This study demonstrated very promising performance of the proposed SAE+OC-SVM model in par with state-of-the art UAD models. Further work will focus on implementing domain adaptation strategies to account for data distribution shift among the different clinical centers, as well as evaluating performance achieved with the FLAIR data only.

## Acknowledgments

This work was granted access to the HPC resources of IDRIS under the allocation 2022-AD011012813R1 made by GENCI. It was partially funded by French program "Investissement d'Avenir" run by the Agence Nationale pour la Recherche (ANR-11-INBS-0006).

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

## Appendix A. Metrics tables for the 3 different hospitals

| Amsterdam | VQ-VAE + Transformer (Pinaya) | AE (Baur) | SAE + multiple OC-SVM (Alaverdyan) | SAE + OC-SVM (Ours) | SAE +OC-SVM +CSF seg (Ours) |
|---|---|---|---|---|---|
| AU ROC | **0.76 ± 0.10** | 0.62 ± 0.08 | 0.62 ± 0.15 | **0.83 ± 0.10** | **0.83 ± 0.07** |
| AU ROC 30 | **0.54 ± 0.16** | 0.34 ± 0.10 | 0.25 ± 0.16 | **0.55 ± 0.21** | **0.65 ± 0.14** |
| AU PRC | **0.084 ± 0.103** | **0.047 ± 0.041** | 0.015 ± 0.018 | **0.099 ± 0.130** | **0.193 ± 0.204** |
| AU PRO | **0.65 ± 0.06** | 0.47 ± 0.06 | 0.41 ± 0.16 | **0.67 ± 0.12** | **0.77 ± 0.07** |
| AU PRO 30 | **0.35 ± 0.09** | 0.15 ± 0.05 | 0.084 ± 0.127 | 0.27 ± 0.20 | 0.43 ± 0.15 |
| ⌈ Dice ⌉ | **0.13 ± 0.11** | **0.10 ± 0.06** | 0.03 ± 0.04 | **0.14 ± 0.15** | **0.25 ± 0.19** |

Table 2: Mean metric on every patient from the Amsterdam hospital for each method. In bold are shown the best model and those for which the statistical difference with the best model for Dunn's test is not significative (p-value ≥ 0.01).

| Singapore | VQ-VAE + Transformer (Pinaya) | AE (Baur) | SAE + multiple OC-SVM (Alaverdyan) | SAE + OC-SVM (Ours) | SAE +OC-SVM +CSF seg (Ours) |
|---|---|---|---|---|---|
| AU ROC | **0.73 ± 0.11** | 0.46 ± 0.03 | 0.51 ± 0.20 | **0.81 ± 0.09** | **0.84 ± 0.09** |
| AU ROC 30 | **0.44 ± 0.15** | 0.13 ± 0.02 | 0.19 ± 0.19 | **0.49 ± 0.20** | **0.64 ± 0.18** |
| AU PRC | **0.074 ± 0.071** | 0.018 ± 0.014 | 0.034 ± 0.045 | **0.090 ± 0.085** | **0.212 ± 0.160** |
| AU PRO | 0.54 ± 0.07 | 0.45 ± 0.04 | 0.47 ± 0.20 | **0.75 ± 0.09** | **0.84 ± 0.05** |
| AU PRO 30 | 0.17 ± 0.07 | 0.10 ± 0.04 | 0.12 ± 0.17 | **0.37 ± 0.16** | **0.55 ± 0.09** |
| ⌈ Dice ⌉ | **0.14 ± 0.11** | 0.04 ± 0.03 | 0.06 ± 0.07 | **0.16 ± 0.12** | **0.27 ± 0.17** |

Table 3: Mean metric on every patient from the Singapore hospital for each method. In bold are shown the best model and those for which the statistical difference with the best model for Dunn's test is not significative (p-value ≥ 0.01).

| Utrecht | VQ-VAE + Transformer (Pinaya) | AE (Baur) | SAE + multiple OC-SVM (Alaverdyan) | SAE + OC-SVM (Ours) | SAE +OC-SVM +CSF seg (Ours) |
|---|---|---|---|---|---|
| AU ROC | 0.58 ± 0.11 | 0.49 ± 0.06 | 0.45 ± 0.16 | **0.76 ± 0.08** | **0.75 ± 0.10** |
| AU ROC 30 | 0.22 ± 0.12 | 0.13 ± 0.04 | 0.12 ± 0.11 | **0.39 ± 0.13** | **0.49 ± 0.12** |
| AU PRC * | 0.038 ± 0.042 | 0.019 ± 0.016 | 0.020 ± 0.017 | 0.062 ± 0.070 | 0.091 ± 0.094 |
| AU PRO | 0.44 ± 0.04 | 0.58 ± 0.08 | 0.41 ± 0.14 | **0.71 ± 0.11** | **0.78 ± 0.05** |
| AU PRO 30 | 0.07 ± 0.02 | 0.20 ± 0.07 | 0.06 ± 0.06 | **0.35 ± 0.15** | **0.46 ± 0.10** |
| ⌈ Dice ⌉ * | 0.07 ± 0.07 | 0.04 ± 0.04 | 0.05 ± 0.04 | 0.11 ± 0.10 | 0.14 ± 0.12 |

Table 4: Mean metric on every patient from the Utrecht hospital for each method. In bold are shown the best model and those for which the statistical difference with the best model for Dunn's test is not significative (p-value ≥ 0.01).
* : Non-significant Kruskal–Wallis test (no best model)

**Appendix B. Databases overview**

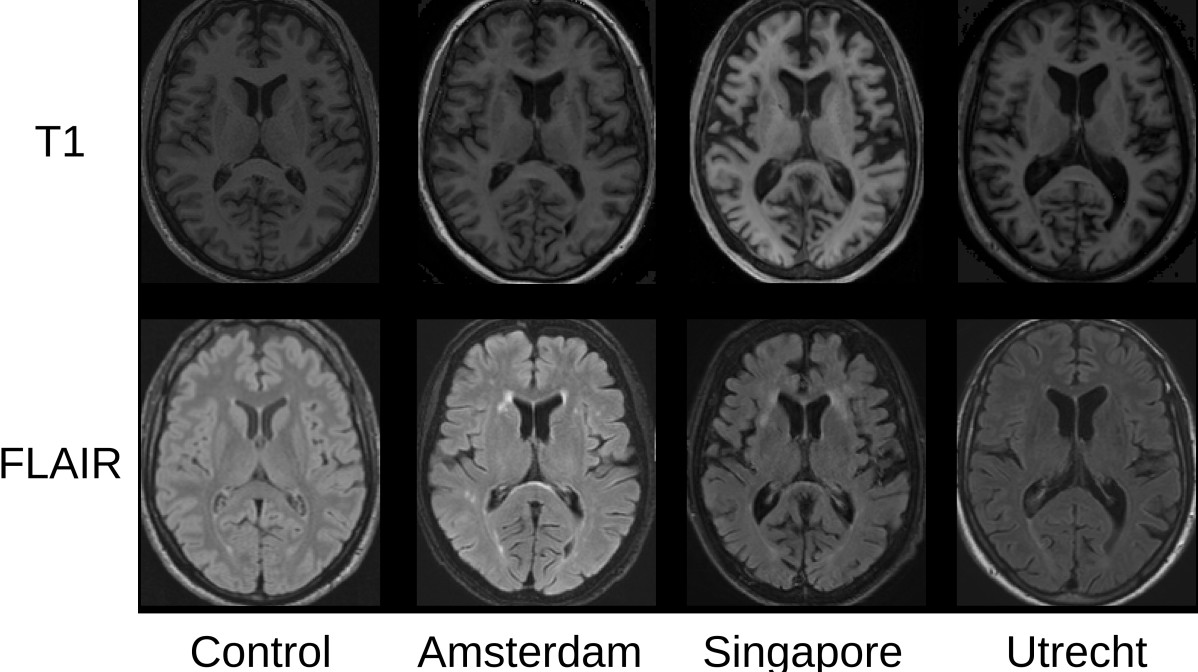

Figure 3: Examples cases of T1 and FLAIR images of the control database and of the 3 hospitals sharing data thhough the WMH database

## Appendix C. Additional results visualization

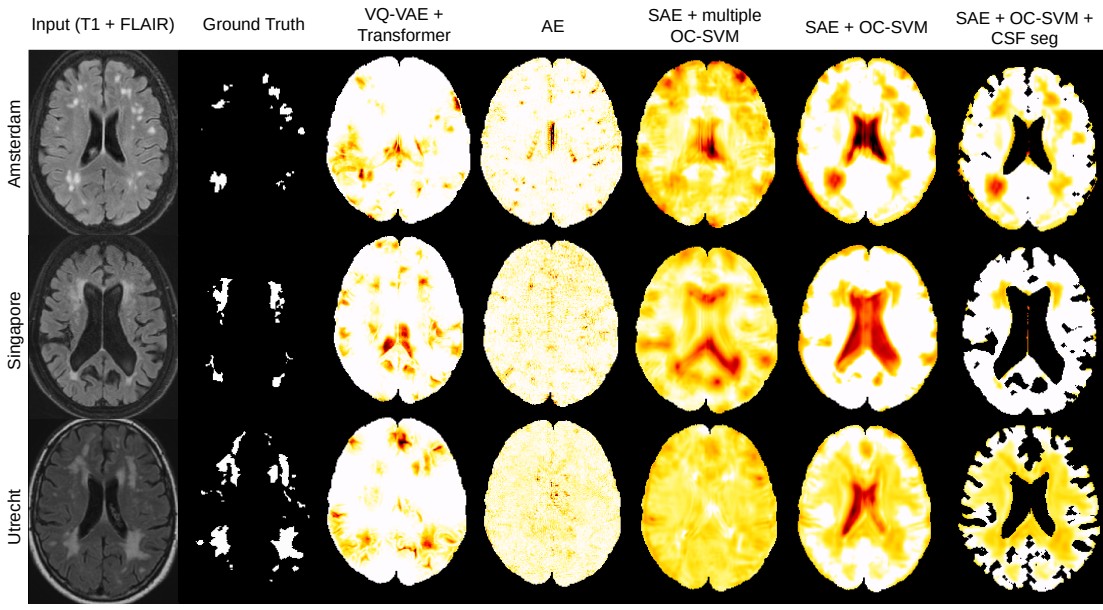

Figure 4: Showcase of the different methods studied on 3 examples (different from figure 2) from the 3 hospitals

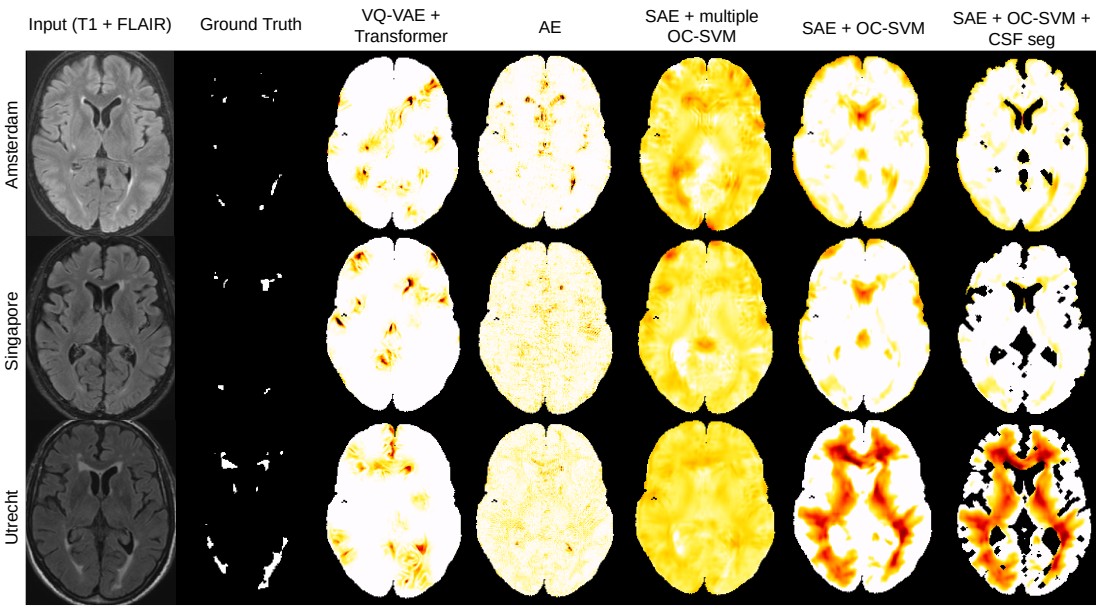

Figure 5: Showcase of the different methods studied on 3 examples (different from figure 2 and 4) from the 3 hospitals

## Appendix D. Influence of the patch size on SAE+OCSVM performances

As to study the influence of the patch size on our method, we ran additional experiments with patch size 9x9, 21x21 and 27x27 to complete the main method with patch size 15x15. Results are reported bellow. Please note that for the 9x9 experiment we had to tweak the auto-encoder by removing the maxpooling and upsampling blocks.

| 3 hospitals | SAE + OCSVM (9x9) | SAE + OCSVM (15x15) | SAE + OCSVM (21x21) | SAE + OCSVM (27x27) |
|---|---|---|---|---|
| AU ROC * | $0.81 \pm 0.14$ | $0.80 \pm 0.09$ | $0.78 \pm 0.09$ | $0.75 \pm 0.12$ |
| AU ROC 30 * | $0.53 \pm 0.25$ | $0.48 \pm 0.20$ | $0.48 \pm 0.16$ | $0.41 \pm 0.19$ |
| AU PRC * | $0.150 \pm 0.171$ | $0.084 \pm 0.099$ | $0.094 \pm 0.098$ | $0.056 \pm 0.062$ |
| AU PRO | $\mathbf{0.70} \pm 0.14$ | $\mathbf{0.71} \pm 0.11$ | $\mathbf{0.71} \pm 0.11$ | $0.61 \pm 0.13$ |
| AU PRO 30 | $\mathbf{0.34} \pm 0.20$ | $\mathbf{0.33} \pm 0.18$ | $\mathbf{0.35} \pm 0.16$ | $0.22 \pm 0.15$ |
| $\lceil$ Dice $\rceil$ * | $0.20 \pm 0.18$ | $0.14 \pm 0.13$ | $0.15 \pm 0.13$ | $0.10 \pm 0.09$ |

Table 5: Mean metric on every patient from the 3 different hospitals for each method. In bold are shown the best model and those for which the statistical difference with the best model for Dunn's test is not significative (p-value $\geq 0.01$).
* : Non-significant Kruskal–Wallis test (no best model)

## Appendix E. Pre-processing and CSF segmentation detailed steps

**Pre-proccessing** Preprocessing of the T1w MR images was performed based on the reference methods implemented in SPM12. The spatial normalization was performed using the unified segmentation algorithm (UniSeg) (Ashburner and Friston, 2005) which includes segmentation of the different tissue types, namely grey matter (GM), white matter (WM) and cerebrospinal fluid (CSF), correction for magnetic field inhomogeneities and spatial normalization to the standard brain template of the Montreal Neurological Institute (MNI). In this work, we used the default parameters for normalization and a voxel size of $1 \times 1 \times 1$ mm. Next, FLAIR image of each subject was rigidly co-registered to its corresponding individual T1w MR image in the native space and then transformed to the MNI space by applying the transformation field produced by the UniSeg algorithm on the T1w image.

The cerebellum and brain stem were excluded from the spatially normalized images. The masking image in the reference MNI space was derived from the Hammersmith maximum probability atlas (Hammers et al., 2003).

On top of that, each image $X$ was intensity-normalized into $X_{norm}$ with : $X_{\mathrm{norm}} = \frac{X - \min(X)}{\max(X) - \min(X)}$.

**CSF segmentation**

As stated in section 2.3, we used the FMRIB's Automated Segmentation Tool (FAST) by (Zhang et al., 2001) to segment the grey and white matter, allowing us to exclude the CSF from the anomaly maps, as we found a high number of false positive in our method belong in these regions.

FAST is here used to provide two CSF segmentation maps, one based on the T1 image and the second based on the T1 and FLAIR images. The union of the two segmentations, after

being masked by a gross brain segmentation to remove the skull, is then processed with some basic mathematical morphology operators, namely : two dilatations followed by two erosions, using a basic cross-shaped structuring element of width 1 voxel. A last erosion on the convex hull of the segmentation is performed to remove a thin outer border of the cortex. Note that this whole processing is done in 3D.

