# OpenReview forum: "One-Class SVM on siamese neural network latent space for Unsupervised Anomaly Detection on brain MRI White Matter Hyperintensities"
_MIDL.io/2023/Conference — MIDL 2023 Poster_

### Official Review · Reviewer_EjzP · 2023-02-03

**Confidence:** 4
**Preliminary Rating:** 3

**Summary:**

The authors propose an unsupervised anomaly detection method for brain lesions, that works on the latent space representations generated by a siamese patch-based autoencoder (SAE, originally proposed by Alaverdyan et al., 2020), in conjunction with a discriminative model – a one-class SVM. They evaluate their methods on MICCAI’s publicly available white matter hyperintensity (WMH) lesions challenge database, and show that their performance is on par with current state-of-the-art methods.

**Strengths:**

- The work seems conceptually and computationally simpler, yet competitive with other current state of the art methods.
- Paper is generally self-contained and fairly easy to follow.
- The basic scaffolding of the work could potentially be explored to solve other related problems in the field, and shows translational promise.


**Weaknesses:**

- Lack of appropriate motivation: While one can appreciate why the authors are interested in finding/detecting lesions, the text does not particularly shed any light on what the drawbacks of methods found currently in the literature are.
- It would help to have the short paragraph at the end of Sec. 2.2 expounded upon earlier, with explanations of what gaps in their methods are, thereby strengthening the need for a new method.
- Why a one-class SVM instead of a more standard multidimensional Gaussian fitting to detect outliers? The latter would potentially require fewer hyperparameters to fit whilst also allowing for precise statements about the probability of there being false positives or negatives using some basic concentration inequalities.


**Deanonymize Review:**

no

**Detailed Comments:**

Major (conceptual) comments:
- Sec 2.1: When the authors say that a couple of images (x_1, x_2) are taken from different patients but from the same location, do they mean that the two images are from the same brain area? Some clarity regarding the same (and to that end, a brief description of the dataset earlier in the text) would be helpful, given that it is non-trivial to have a 1:1 mapping between the brains of two subjects.
- How is the appropriate value for $\alpha$ determined? Assuming it is a hyper-parameter, how was it estimated? Were there any appreciable trends in the quality of the latent representations generated when varying the same?
- Sec 2.2: “... assuming that the lesion area represents a negligible fraction of patient images” How would the method extend to the case where the number of lesion images no longer satisfies the relation n<<m? When does this relationship hold? (I.e., how big can the ratio n/m get?) While it seems like a reasonable assumption to make given the dataset, would it necessarily be true if the method were to be applied in a clinical setting (especially since imaging more of the brain would mean spending more time and money for doing so)?
- Why are the numbers for Alaverdyan et al., 2020 not included, given that’s the method that is most comparable to the one proposed by the authors?

Minor (stylistic) comments:
- Typo, Sec. 2.1: “The siamese auto-encoder is composed of two auto-encoders with shared weight” – Change to “weights”
- Fig. 1: The pipeline’s diagram shows just the reconstruction term as the objective, while the text states there also exists a term that penalizes the cosine dissimilarity between the latent encodings of images drawn from the same location. Would be helpful to have both parts of the objective function included in Fig.1 (with appropriate arrows added, of course).
- The objective function as stated is almost lost; Would be nice to have it highlighted and/or center aligned in the text with some gap between it and the text that follows it.


**Paper Type:**

validation/application paper

**Questions To Address In The Rebuttal:**

- Rewrites so that the need for the method is better motivated, and also why it might be more efficient (if not necessarily) than other SotA methods w.r.t. compute and data.
- Any scope for quantitatively ensuring you don’t cross a target false positive/negative rate?
- It would be nice to have a sense for what else the learnt representations from the SAE could be used for. Do they provide any intuitions or point to promising avenues of future work?

---

### Official Review · Reviewer_jkJ3 · 2023-02-06

**Confidence:** 4
**Preliminary Rating:** 4
**Recommendation:** Poster

**Summary:**

The paper proposes an unsupervised anomaly detection (UAD) method for multi-modality neuroimaging. The method uses a siamese patch-based auto-encoder to construct a latent space and performs outlier detection with a One-Class SVM training paradigm. The model is evaluated on a public database (White Matter Hyperintensities challenge) and performs comparably to the two best-performing state-of-the-art methods.

**Strengths:**

1. The paper presents a new Unsupervised Anomaly Detection (UAD) approach that identifies outliers by using a One-Class Support Vector Machine (OC-SVM) trained on healthy-controlled patches. The anomaly detection procedure does not necessitate additional training and has been thoroughly validated through a sufficient number of experiments, demonstrating the effectiveness of the proposed methodology.

2. This work employs a straightforward framework and utilizes an open-source dataset, facilitating ease of replication.

**Weaknesses:**

1. The authors of the paper note that poor registration performance results in a high incidence of false positives in the cerebrospinal fluid (CSF). Instead of addressing the registration issue directly, the authors elect to segment the gray and white matter and exclude the CSF from the anomaly maps. The reviewer proposes that the authors explicitly list their comprehensive pre-processing steps, including but not limited to registration, normalization, and histogram equalization, to assist readers in their analysis and provide them with an understanding of the issue.

2. Despite the simplicity of the framework employed in this study, the requirement of a substantial quantity of training data for unsupervised learning, in addition to additional constraints such as the prerequisite of CSF segmentation and the need for paired training data, amplify the difficulty of implementing this network for other datasets or tasks.

**Deanonymize Review:**

no

**Detailed Comments:**

1. The utilization of paired data increases the challenge of applying this network to other datasets or tasks. The reviewer expresses curiosity about the potential for achieving similar UAD performance by using (x, x') as input, where x' is an augmented version of a single patch x, and altering the cosine loss to a patch-based contrastive loss, when working with unpaired datasets.

3. The visual evaluation presented in Figure 2 does not provide clear evidence of the superiority of the proposed method over state-of-the-art methods. The reviewer recommends that the authors present results for more samples to facilitate a more thorough comparison and analysis.

**Paper Type:**

methodological development

**Questions To Address In The Rebuttal:**

The reviewer suggests the authors consider both the mentioned Weakness and Comments, and address the following two issues:
1) The reviewer proposes that the authors explicitly list their comprehensive pre-processing steps.
2) Add qualitative evaluation results for more samples.

---

### Official Review · Reviewer_HFc4 · 2023-02-07

**Confidence:** 4
**Preliminary Rating:** 2

**Summary:**

The authors proposed a novel method for unsupervised anomaly detection (UAD) on neuroimaging. Currently, most UAD models are auto-encoder based and rely on reconstruction errors to localise anomalies, while they have limited capability in detecting subtle anomalies. The authors follow the patch-based method introduced by Alaverdyan et al (2020), and train a siamese autoencoder (SAE) to reconstruct 2D patches of healthy images. Specifically, SAE takes a patch at the same location but from different subjects and forces them to have similar latent representations. To detect the anomalies, one-class SVM (OC-SVM) is used. The method is evaluated on the WMH segmentation challenge dataset and compared to previous methods.


**Strengths:**

UAD has been a new and interesting topic, and can make the localization of anomalies more efficient at a lower cost of annotations. The paper is overall well organized and compared to previous works.

**Weaknesses:**

- The proposed method is built upon the work by Alaverdyan et al (2020), could the authors specify the major contributions and novelties introduced by this work?
- To draw patches at the same location from different subjects, it’s necessary to ensure the subjects have similar resolution in each direction and aligned to a similar pose, is there any pre-processing on the WMH dataset to ensure this?
- In Fig.2, could the authors show detected anomalies more clearly? The visualization makes it hard to compare among the methods as well as to the ground truths.
- As this work is inspired by the work by Alaverdyan et al (2020), is it possible to add their method in the comparison?
- What is the patch size used in this work and how does the patch size affect the performance of the model?
- Minor issue: in page 3, in the definition of $L_{SAE} (x_1, x_2)$, the subscript $t$ is not used in the equation, there seems to be a mistake in the definition of the reconstruction loss.

**Deanonymize Review:**

no

**Paper Type:**

both

**Questions To Address In The Rebuttal:**

The paper addresses an important issue in UAD, as current UAD have limited capability in detecting subtle anomalies. The authors give a relative thorough overview of related works on this topic, and proposed a method inspired by the patch-based method by Alaverdyan et al (2020).

- The proposed method seems very similar to the method by Alaverdyan et al, could the authors further explain the major contributions and critical modifications in this work?
- Improve the visualisation quality and fix minor issues with proofreading.
- Is there specific pre-processing required to ensure patches extracted from different subjects are truly from a similar location?
- Discuss the effect of patch size on the model performance and how to select a suitable patch size for the model.

---

### Meta-Review · Area_Chair_sQZd · 2023-02-25

**Recommendation:** Accept (Poster)
**Confidence:** 5

**Metareview:**

The authors propose a new approach for unsupervised anomaly detection. However there were concerns raised by reviewers around: 1) novelty - the authors improvements over Alaverdyan et al. (2020) do not seem to be significant; 2) clarity of writing - the motivation and novelty are not clearly described. The rebuttal has done a good job in addressing these concerns and as a result I recommend accepting the paper.